# Impact of Chronic Kidney Disease on Chronic Total Occlusion Revascularization Outcomes: A Meta-Analysis

**DOI:** 10.3390/jcm10030440

**Published:** 2021-01-23

**Authors:** Wei-Chieh Lee, Po-Jui Wu, Chih-Yuan Fang, Huang-Chung Chen, Chiung-Jen Wu, Hsiu-Yu Fang

**Affiliations:** 1Division of Cardiology, Department of Internal Medicine, Kaohsiung Chang Gung Memorial Hospital, Chang Gung University College of Medicine, Kaohsiung 83301, Taiwan; sky1021@cgmh.org.tw (P.-J.W.); cyfang@seed.net.tw (C.-Y.F.); inq39@yahoo.com.tw (H.-C.C.); cvcjwu@cloud.cgmh.org.tw (C.-J.W.); ast42aiu@hotmail.com (H.-Y.F.); 2Institute of Clinical Medicine, College of Medicine, National Cheng Kung University, Tainan 83301, Taiwan

**Keywords:** chronic kidney disease, chronic total occlusion, revascularization, percutaneous coronary intervention

## Abstract

Objectives: To examine the impact of revascularization and associated clinical outcomes of chronic kidney disease (CKD) chronic total occlusion (CTO) and non-CKD CTO groups. Background: The influence of CKD on clinical outcomes after percutaneous coronary intervention (PCI) for CTO lesions is unknown, and there is no systemic review of this topic to date. Methods: We searched the PubMed, Embase, ProQuest, ScienceDirect, Cochrane Library, ClinicalKey, Web of Science, and ClinicalTrials Databases for articles published between 1 January 2010 and 31 March 2020. CKD was defined as estimated glomerular filtration rate of <60 mL/min/1.73 m^2^ according to the Modification of Diet in Renal Disease formula. Data included demographics, lesion distributions, incidence of contrast-induced nephropathy (CIN), acute kidney injury (AKI), procedural success rate, mortality, and target lesion revascularization (TLR)/target vessel revascularization (TVR). Results: Six studies were ultimately included in this systematic review. A high prevalence (25.5%; range, 19.6–37.9%) of CKD was noted in the CTO population. In the non-CKD group, outcomes were better: less incidence of CIN or AKI (odds ratio (OR), 2.860; 95% confidence interval (CI), 1.775–4.608), higher procedural success rate (OR, 1.382; 95% CI, 1.036–1.843), and lower long-term mortality (OR, 4.502; 95% CI, 3.561–5.693). The incidence of TLR/TVR (OR, 1.118; 95% CI, 0.888–1.407) did not differ between groups. Conclusions: In the CKD CTO PCI population, a lower procedural success rate, a higher incidence of CIN or AKI, and higher in-hospital and long-term mortality rate were noted due to more complex lesions and more comorbidities. However, the incidence of TLR/TVR did not differ between groups.

## 1. Introduction

Chronic kidney disease (CKD) is common in patients presenting with symptomatic coronary chronic total occlusion (CTO), with a reported prevalence of 10% [1]. However, the comparative safety and effectiveness of percutaneous coronary intervention (PCI) in these patients have not been thoroughly described to date. CTO PCI is associated with improved symptoms, health status, and left ventricular performance and increased exercise tolerance [2,3,4]. CTO PCI remains challenging for interventionists and frequently involves longer procedures and greater iodinated contrast exposure; thus, patients with CKD might experience a higher incidence of contrast nephropathy [5,6]. Patients with CKD patients have multiple comorbidities and increased coronary lesion complexity that could adversely affect PCI outcomes. The influence of CKD on clinical outcomes after PCI for CTO lesions is unknown and mostly limited by a rather small patient number [7,8,9,10,11,12]. Furthermore, there are no systematic reviews on this topic to date. Here, we compared the impact of revascularization by PCI and the associated clinical outcomes between CKD CTO and non-CKD CTO groups.

## 2. Methods

This systematic review followed the Preferred Reporting Items for Systematic Reviews and Meta-analyses reporting guidelines, and the study protocol adhered to the requirements of the institutional review board of our Hospital.

### 2.1. Search Strategies, Trial Selection, and Quality Assessment and Data Extraction

Two cardiologists (Po-Jui Wu and Wei-Chieh Lee) separately performed a systematic literature search of the PubMed, Embase, ProQuest, ScienceDirect, Cochrane Library, ClinicalKey, Web of Science, and ClinicalTrials.gov databases for articles published between 1 January 2010 and 31 March 2020. The databases were searched for relevant studies without language restrictions using the key terms “chronic kidney disease” (CKD), and “coronary chronic total occlusion” (CTO), and “percutaneous coronary intervention” (PCI). Disagreements were resolved by a third reviewer (Hsiu-Yu Fang). Only randomized controlled trials and observational studies comparing the clinical outcomes of CTO revascularization were included in the present meta-analysis. The inclusion criteria were human studies with a parallel design and comparisons of clinical outcomes between patients with or without CKD after revascularization. The exclusion criteria were case reports or series, animal studies, review articles, and conference abstract. We did not set any language limitations to increase the number of eligible articles. Figure 1 shows the literature search and screening protocol. However, only 4 studies clearly provided the incidence of contrast-induced acute kidney injury after intervention for CTO in 6 selected studies. 

### 2.2. Definitions

CTO was defined as coronary obstruction with Thrombolysis In Myocardial Infarction (TIMI) flow grade 0 and an estimated occlusion duration of ≥3 months on previous coronary angiography or clinical definition (a history of angina or history of myocardial infarction in the same territory) [13,14]. CKD was defined as estimated glomerular filtration rate (eGFR) of <60 mL/min/1.73 m^2^ according to the Modification of Diet in Renal Disease formula [15]. Procedural success was defined as a final TIMI flow grade of at least 2 and a residual stenosis of no greater than 30% after stent implantation or a TIMI flow grade of 3 with ≤50% residual stenosis and no major adverse cardiovascular events (Table 1). Contrast-induced nephropathy (CIN) and acute kidney injury (AKI) were defined as an absolute increase in serum creatinine of at least 0.5 mg/dL or a relative increase of at least 25% that occurred within 48–72 h after PCI [16]. Renal replacement therapy (RRT) was defined as the need for dialysis after PCI or for a short-term period. Long-term outcome was defined as the outcomes after discharge during the follow-up period. Target lesion revascularization (TLR)/target vessel revascularization (TVR) was defined as prior treated lesion or vessel that needs revascularization due to restenosis during the follow-up period. 

### 2.3. Statistical Analysis

The frequency of each evaluated outcome was abstracted from each study and the data are presented as cumulative rates. To assess heterogeneity across trials, we used the chi-squared (*p* ≤ 0.1 considered significant) and *I^2^* statistic (25%, 50%, and 75% correlated with low, moderate, and high heterogeneity, respectively) to examine each outcome.

## 3. Results

### 3.1. Characteristics of the Included Studies

Six cohort studies met the inclusion criteria. The study selection process is shown in Figure 1. The study design and participants’ characteristics are described in Table 2. A total of 9863 participants (mean age, 65 years; 15.6% female) were included. The prevalence of CKD was 26.2%. The study period, comparison, follow-up time, and the prevalence of CKD data are shown in Table 2. 

### 3.2. Patient Demographics and CTO Target Vessel

Table 3 describes the basic demographics and CTO target vessels of the study patients. The CKD CTO group had a higher mean age (70.6 ± 8.9 years vs. 63.9 ± 9.4 years, *p* < 0.001) and a higher prevalence of female patients (20.4% vs. 14.1%, *p* < 0.001). The CKD CTO group also had a higher prevalence of comorbidities, including diabetes mellitus, heart failure, and previous coronary artery bypass grafting. Lower mean left ventricular ejection fraction and eGFR values were also noted. The rates of left descending anterior and right coronary artery CTO were higher in both groups. Mean Japan CTO (J-CTO) score was significantly higher in the CKD CTO group (2.4 ± 1.3 vs. 2.2 ± 1.3, *p* < 0.001). Longer mean procedure and fluoroscopy times were noted in the CKD CTO group. Significantly less contrast volume was used in CKD CTO group (CKD vs. non-CKD, 255.4 ± 106.7 mL vs. 285.0 ± 116.3 mL, *p* < 0.001). The method used regarding retrograde wire escalation and retrograde dissection and reentry did not differ between CKD and non-CKD groups.

### 3.3. Pooled ORs of CIN, AKI, or RRT after PCI

Four studies mentioned the incidence of CIN or AKI and the need for RRT. The overall odds ratio (OR) of the incidence of CIN or AKI in the CKD CTO versus non-CKD CTO arm was 2.860 (95% CI, 1.775–4.608) (Figure 2A), with low heterogeneity (Cochran Q, 5.788; *df*, 3; *I*^2^, 48.082%; *p* = 0.123) and insignificant publication bias via Egger regression (*t*, 2.407; *df*, 2; *p* = 0.138). Only one study discussed the incidence of RRT. The OR of the incidence of RRT in the CKD CTO versus non-CKD CTO arm was 8.341 (95% CI, 0.752–92.505) (Figure 2B).

### 3.4. Pooled ORs of Procedural Success Rate between Groups

According to five studies, the overall OR of the procedural success rate in the CKD CTO versus non-CKD CTO arm was 1.382 (95% CI, 1.036-1.843) (Figure 3), with moderate heterogeneity (Cochran Q, 12.971; *df*, 4; *I*^2^, 69.161%; *p* = 0.011) and insignificant publication bias via Egger regression (*t*, 0.129; *df*, 43; *p* = 0.905). 

### 3.5. Pooled ORs of in-Hospital and Long-Term Mortality Rates

Two studies compared in-hospital mortality rates between groups. The overall OR of the incidence of in-hospital mortality in the CKD CTO versus non-CKD CTO arm was 4.822 (95% CI, 2.332–9.973) (Figure 4A), showing insignificant heterogeneity (Cochran Q, 0.576; *df*, 1; *I*^2^, 0%; *p* = 0.448). 

Three studies mentioned the incidence of long-term mortality. The overall OR of long-term mortality in the CKD CTO versus non-CKD CTO arm was 4.502 (95% CI, 3.561–5.693) (Figure 4B), showing insignificant heterogeneity (Cochran Q, 0.047; *df*, 2; *I*^2^, 0%; *p* = 0.977) and insignificant publication bias via Egger regression (*t*, 0.858; *df*, 1; *p* = 0.548).

### 3.6. Pooled ORs of TLR or TVR between Groups

According to three studies, the overall OR of the incidence of TLR/TVR in the CKD CTO versus non-CKD CTO arm was 1.118 (95% CI, 0.888–1.407) (Figure 5), with low heterogeneity (Cochran Q, 3.256; *df*, 2; *I*^2^, 38.574%; *p* = 0.196) and insignificant publication bias via Egger regression (*t*, 0.139; *df*, 1; *p* = 0.912). 

## 4. Discussion

CKD is increasingly recognized as a global public health problem, and epidemiological investigations have confirmed the trend of increasing morbidity rates in CKD populations. Six studies of CTO PCI populations reported a high prevalence (mean, 25.5%; range, 19.6–37.9%) of CKD [7,8,9,10,11,12]. Older age, higher prevalence of female sex, more comorbidities, and poorer left ventricular performance were noted in the CKD CTO group. A higher mean J-CTO score was noted in the CKD CTO group. Therefore, increased coronary lesion complexity was noted in this population that required longer procedure and fluoroscopy times for PCI. Less contrast volume was used to prevent CIN or AKI. 

Most studies included in this analysis showed a significant trend of CIN and AKI in the CKD CTO group [7,9,11,12]. Azzalini et al. reported that CIN-related AKI was more prevalent in the CKD CTO group, but dialysis was infrequently required [9]. Several preventive methods for CIN have been described in the literature, but most remain controversial [17]. Minimization of contrast media volume and achievement of meticulous patient hydration are important for preventing CIN. Therefore, significantly less contrast volume was used in the CTO CKD group than in the non-CKD CTO group. The four studies [7,10,11,12] had a higher prevalence of female patients. Female patients reportedly experience more in-hospital complications and CIN after CTO PCI as well [18].

In-hospital and long-term mortality rates were favorable in the non-CKD group because of younger mean patient age, fewer comorbidities, and higher procedural success rates. In low-risk or general populations, a lower eGFR is reportedly an important predictor of all-cause mortality [19,20]. CKD also has an important impact on mortality in CTO populations [10,11]. Successful CTO recanalization still could improve survival irrespective of renal function [10]. In one large registry, a higher incidence of TLR was noted in the CKD group, especially in patients with severe CKD or who required dialysis [21]. In our analysis, the target lesion failure rate was similar between the CKD CTO and non-CKD CTO groups. 

To the best of our knowledge, no systematic review and meta-analysis of the impact of CKD on the clinical outcomes of CTO populations was reported in the past 5 years. In our analysis, poorer clinical condition and greater coronary artery complexity were noted in the CKD CTO group. In addition, poorer clinical outcomes including a higher incidence of CIN, AKI, and mortality were presented. A lower procedural success rate was also noted. However, the need for dialysis and the incidence of TLR/TVR did not differ between groups. 

### Limitations

This study had several limitations; thus, its findings must be considered cautiously. First, all studies were observational cohort studies. However, a total of 8232 participants were included from six studies. Second, the analysis of procedural success rate, which differed among studies, showed moderate heterogeneity (Table 1). Third, not all studies provided the level of creatinine and the incidence of CIN or AKI. Therefore, the incidence of CIN and AKI may be underreported. Fourth, no data about outcomes in successful versus unsuccessful PCI in such comparison were presented. However, the present study still provides important information about CTO PCI in CKD populations.

## 5. Conclusions

In CKD CTO PCI population, a lower procedural success rate, a higher incidence of CIN or AKI, and higher in-hospital and long-term mortality rate were noted due to more complex lesions and more comorbidities. However, the incidence of TLR/TVR did not differ between groups.

## Figures and Tables

**Figure 1 jcm-10-00440-f001:**
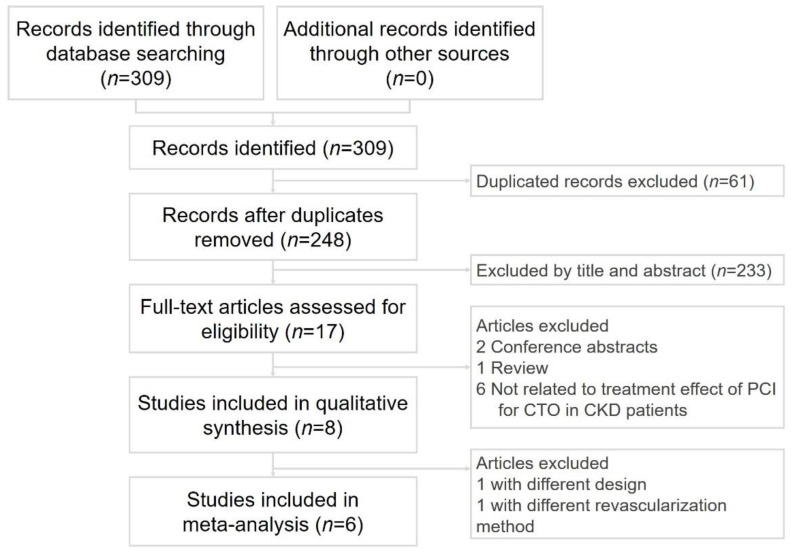
Flowchart of the selection strategy and inclusion and exclusion criteria for this meta-analysis. PCI, percutaneous coronary intervention; CTO, chronic total occlusion; CKD, chronic kidney disease.

**Figure 2 jcm-10-00440-f002:**
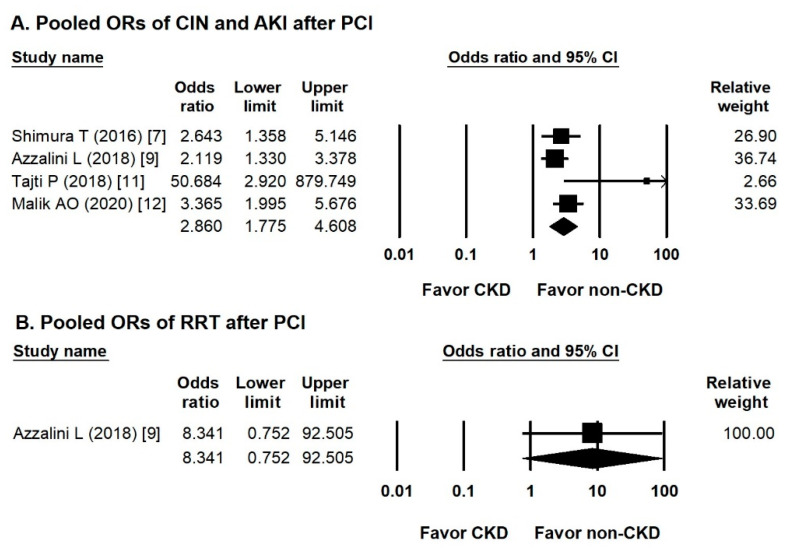
Forest plots of the renal events. (**A**) Forest plots of the incidence of contrast-induced nephropathy or acute kidney injury between the CKD and non-CKD groups from four studies. (**B**) Forest plots of the incidence of renal replacement therapy between the CKD and non-CKD groups from one study.

**Figure 3 jcm-10-00440-f003:**
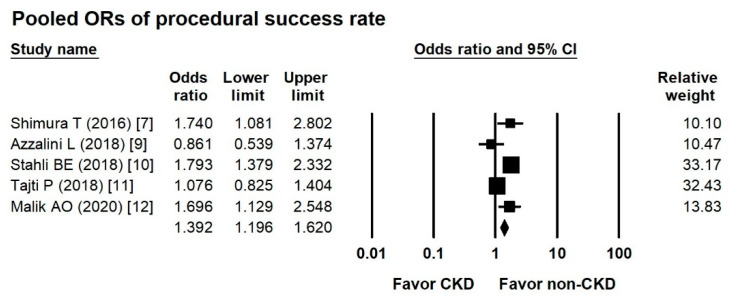
Forest plots of the procedural success rate between the CKD and non-CKD groups from six studies.

**Figure 4 jcm-10-00440-f004:**
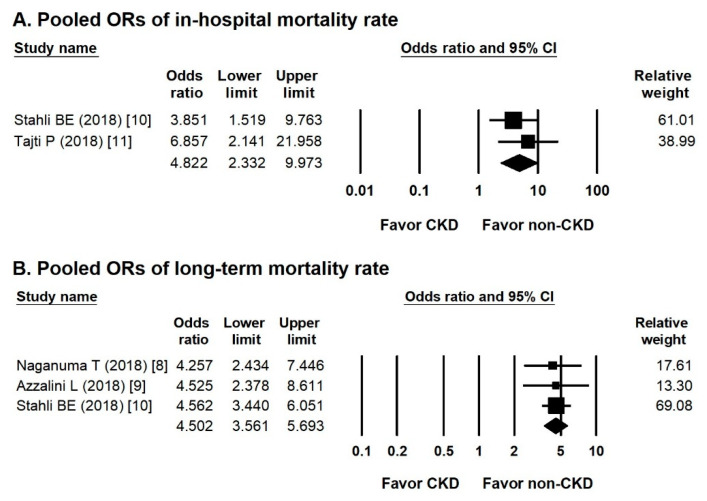
Forest plots of in-hospital and long-term mortality rate. (**A**) Forest plots of the incidence of in-hospital mortality between the CKD and non-CKD group from two studies. (**B**) Forest plots of the incidence of long-term mortality between the CKD and non-CKD groups from four studies.

**Figure 5 jcm-10-00440-f005:**
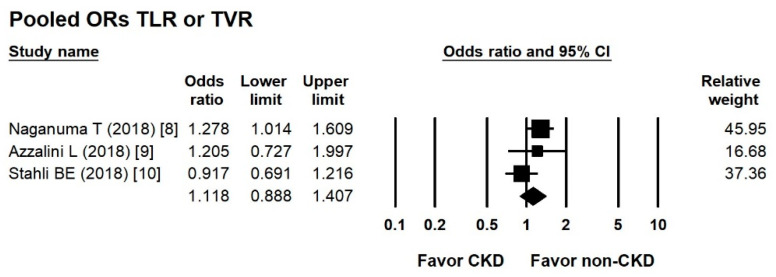
Forest plots of the incidence of target lesion revascularization and target vessel revascularization in the CKD and non-CKD groups from four studies.

**Table 1 jcm-10-00440-t001:** Characteristics of the 7 included studies.

First Author (Year)	The Definition of Procedural Success	Procedural Successful Rate (CKD vs. Non-CKD)
Shimura T. (2016) [7]	TIMI flow grade 3 and a residual stenosis of ≤50% and without MACE	83.1% vs. 89.5%
Naganuma T. (2018) [8]	N/A	N/A
Azzalini L. (2018) [9]	TIMI flow grade = 3 and a residual stenosis < 30%	79% vs. 86%
Stahli B.E. (2018) [10]	TIMI flow grade = 3 and a residual stenosis < 30%	75.8% vs. 84.9%
Tajti P. (2018) [11]	TIMI flow grade = 3 and a residual stenosis < 30%	83.0% vs. 84.0
Malik A.O. (2020) [12]	TIMI flow grade ≥ 2 and a residual stenosis < 50% and no side branch occlusion	81.8% vs. 88.4%

Abbreviation: CKD: chronic kidney disease; TIMI, Thrombolysis In Myocardial Infarction.

**Table 2 jcm-10-00440-t002:** Characteristics of the 6 included studies.

First Author (Year)	Patients Number (Female %)	Age (Years)	Study Period	The Prevalence of CKD (%)	Comparison of Groups	Follow-Up
Shimura T. (2016) [7]	739 (16.6)	65.9 ± 10.8	January 2006–December 2013	24.0	CKD PCI (177) vs. non-CKD PCI (562)	4.3 ± 2.4 years
Naganuma T. (2018) [8]	1463 (17.4)	66.6 ± 10.4	August 2004–December 2014	37.9	CKD PCI (555) vs. non-CKD PCI (908)	4.6 (2.3–6.7) years
Azzalini L. (2018) [9]	1092 (10.6)	64.9 ± 10.3	July 2011–June 2017	19.6	CKD PCI (214) vs. non-CKD PCI (878)	1.3 (0.9–3.1) years
Stahli B.E. (2018) [10]	2002 (16.6)	65.9 ± 8.6	January 2005–December 2013	20.9	CKD PCI (418) vs. non-CKD PCI (1584)	2.6 (1.1–3.1) years
Tajti P. (2018) [11]	1979 (14.4)	65.0 ± 10.0	May 2012–November 2017	27.0	CKD PCI (535) vs. non-CKD PCI (1444)	In-hospital
Malik A.O. (2020) [12]	957 (19.4)	65.3 ± 10.3	January 2014–July 2015	23.6	CKD PCI (225) vs. non-CKD PCI (732)	1 year

Abbreviation: CKD: chronic kidney disease; PCI, percutaneous coronary intervention.

**Table 3 jcm-10-00440-t003:** Patients demographics and CTO target vessel.

	CKD	Non-CKD	*p* Value
Age (years)	70.6 ± 8.9 (2124)	63.9 ± 9.4 (6108)	<0.001
Female sex (%)	20.4 (434)	14.1 (862)	<0.001
Diabetes mellitus (%)	46.3 (983)	33.3 (2036)	<0.001
Heart failure (%)	39.1 (545)	22.1 (1025)	<0.001
LVEF (%)	49.3 ± 13.8 (1529)	52.8 ± 12.6 (3962)	<0.001
Previous CABG (%)	30.3 (475)	19.9 (1033)	<0.001
eGFR (mL/min/1.73m^2^)	40.7 ± 17.4 (2124)	85.1 ± 16.3 (6108)	<0.001
J-CTO score	2.4 ± 1.3 (995)	2.2 ± 1.3 (3094)	<0.001
Procedure time (mins)	139.2 ± 69.3 (1172)	130.9 ± 71.9 (3656)	<0.001
Fluoroscopy time (mins)	43.0 ± 27.7 (1413)	39.1 ± 26.7 (4678)	<0.001
Contrast volume (mL)	255.4 ± 106.7 (1590)	285.0 ± 116.3 (5240)	<0.001
Retrograde wire escalation and retrograde dissection and reentry (%)	30.0 (395)	29.0 (1358)	0.701

Data are expressed as mean ± standard deviation or as number (percentage). Abbreviation: CTO: chronic total occlusion; CKD: chronic kidney disease; CABG: coronary artery bypass graft; eGFR: estimated Glomerular filtration rate; J-CTO: Japan CTO.

## Data Availability

No new data were created in this study. Data sharing is not applicable to this article.

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
