# Peer review of "Impact of Chronic Kidney Disease on Chronic Total Occlusion Revascularization Outcomes: A Meta-Analysis"

_jcm, 2021, doi:10.3390/jcm10030440_

Round 1

Reviewer 1 Report

Dear authors,

thank for for performing a meta-analysis on CKD and CTO-PCI outcomes.

Please find below my comments:

  1. in the abstract the definition of CKD needs to be added
  2. in the abstract: favorable outcomes of CIN or AKI....were noted in the non-CKD group. The formulation is confusing: would suggest to start as follows: In the non-CKD group, outcomes were better: higher success on intervention (OR..), less...
  3. In the review proces, one should know how the authors of the different studies have confirmed AKI after PCI. In principle creatinine levels should have been checked at least once and preferably twice. However, most studies do not provide creatinine levels in all of there patients - this is a major limitation and result in under-reporting of AKI after PCI. It should be mentioned in methods (and if unclear, insufficient mentioned in limitations)
  4. Table 2: FU: use one definition: or days, or months or years but not all of them
  5. What is long term outcome? After 30 days? Please define.
  6. Ref 9 = CABG + PCI: only the PCI population data should be used. 
  7. Line 147-151 = 163-167, skip the second one
  8. Line 169: higher(?) procedural success
  9. A table should be added comparing age, sex, LV function, J-CTO, contrast use in non-CKD versus CKD. It is mentioned in discussion, but it is not reported in results. 
  10. Was there a difference in techniques used in non-CKD vs CKD (more retro, ADR in CKD?)
  11. TLR: within which period of time? 

Author Response

Specific responses to the first reviewer’s comments:

Reviewer #1:

Comment 1: in the abstract the definition of CKD needs to be added

Responses: We have revised our manuscript according to this comment and revised one paragraph in Abstract on page 1, paragraph 1, lines 16-17, of the revised manuscript:

“CKD was defined as estimated glomerular filtration rate of < 60 mL/min/1.73 m2 according to the Modification of Diet in Renal Disease formula.”

Comment 2: in the abstract: favorable outcomes of CIN or AKI....were noted in the non-CKD group. The formulation is confusing: would suggest to start as follows: In the non-CKD group, outcomes were better: higher success on intervention (OR..), less...

Responses: We have revised our manuscript according to this comment and revised one paragraph in Abstract on page 1, paragraph 1, lines 21-24, of the revised manuscript:

“In the non-CKD group, outcomes were better: less incidence of CIN or AKI (odds ratio [OR], 2.860; 95% confidence interval (CI), 1.775-4.608), higher procedural success rate (OR, 1.433; 95% CI, 1.113-1.844), and lower long-term mortality (OR, 4.483; 95% CI, 3.665-5.483).”

Comment 3: In the review proces, one should know how the authors of the different studies have confirmed AKI after PCI. In principle creatinine levels should have been checked at least once and preferably twice. However, most studies do not provide creatinine levels in all of there patients - this is a major limitation and result in under-reporting of AKI after PCI. It should be mentioned in methods (and if unclear, insufficient mentioned in limitations)

Responses: We have revised our manuscript according to this comment and revised one paragraph in Method on page 2, paragraph 3, lines 72-74, and in limitations on page 7, paragraph 1, lines 217-219, of the revised manuscript:

“However, only 4 studies clearly provided the incidence of contrast-induced acute kidney injury after intervention for CTO in selected 6 studies.” 

“Third, not all studies provided the level of creatinine and the incidence of CIN or AKI. Therefore, the incidence of CIN and AKI may be underreported.”

Comment 4: Table 2: FU: use one definition: or days, or months or years but not all of them

Responses: We have revised Table 2 according to this comment and used years in Table 2.

Comment 5: What is long term outcome? After 30 days? Please define.

Responses: We have revised our manuscript according to this comment and revised one paragraph in Definitions on page 3, paragraph 1, lines 90-91, of the revised manuscript:

“Long-term outcome was defined as the outcomes after discharge during follow-up period.”

Comment 6: Ref 9 = CABG + PCI: only the PCI population data should be used. 

Responses: We deleted this reference in revised manuscript and tables according to this comment. We also modified the number of references.

Comment 7: Line 147-151 = 163-167, skip the second one

Responses: We deleted this paragraph in revised manuscript according to this comment.

Comment 8: Line 169: higher(?) procedural success

Responses: We have revised our manuscript according to this comment and revised one paragraph in Discussion on page 7, paragraph 4, line 197, of the revised manuscript:

“In-hospital and long-term mortality rates were favorable in the non-CKD group because of younger mean patient age, fewer comorbidities, and higher procedural success rates.”

Comment 9: A table should be added comparing age, sex, LV function, J-CTO, contrast use in non-CKD versus CKD. It is mentioned in discussion, but it is not reported in results. 

Responses: We added Table 3 according to this comment.

Comment 10: Was there a difference in techniques used in non-CKD vs CKD (more retro, ADR in CKD?)

Responses: We have revised our manuscript according to this comment and revised one paragraph in Results on page 4, paragraph 2, lines 120-122, of the revised manuscript:

“The used method about retrograde wire escalation and retrograde dissection and reentry did not differ between CKD and non-CKD groups.”

Comment 11: TLR: within which period of time? 

Responses: We have revised our manuscript according to this comment and revised one paragraph in Definitions on page 3, paragraph 1, lines 91-93, of the revised manuscript:

“Target lesion revascularization (TLR)/target vessel revascularization (TVR) was defined as prior treated lesion or vessel which need revascularization due to restenosis during follow-up period.”

Thank you for your constructive and valuable comments.

Reviewer 2 Report

The authors performed a meta-analysis of 7 cohort studies of patients with chronic total oclussions (CTO). Outcomes were assessed between patients with and without chronic kidney disease (CKD). Results showed lower procedural success rate, higher incidence of CIN or AKI, and higher mortality rates. Clinical restenosis rates were comparable between groups.

Main comments:

Given the paucity of (randomized) data in this field (CTO and CKD), results of this meta-analysis are welcome. There are few issues to be addressed by the authors:

  • The authors stated that CKD patients had worse baseline clinical characteristics. However, these are not presented in the results section. They are mentioned only in the first paragraph of the discussion. As this is one of the key elemets to understand the results (and the lower success rate), I recommend adding a paragraph /table on baseline characteristics in the results section. As no propensity analysis or multivariate analysis have been performed one can not rule out the precise reason (CKD vs other baseline condition) for worse outcomes (i.e. mortality).
  • In the abstract, it is also mentioned the more complex lesions treated in the CKD group but no data are presented in this regard.
  • The last sentence of the abstract is not a conclusion, rather an objective of the study. Please rephrase the conclusions.
  • Data on outcomes in successful vs unsuccessful PCI are not presented. Please add this in the limitations section. 

Author Response

Specific responses to the second reviewer’s comments:

Reviewer #2:

The authors performed a meta-analysis of 7 cohort studies of patients with chronic total oclussions (CTO). Outcomes were assessed between patients with and without chronic kidney disease (CKD). Results showed lower procedural success rate, higher incidence of CIN or AKI, and higher mortality rates. Clinical restenosis rates were comparable between groups.

Comment 1: The authors stated that CKD patients had worse baseline clinical characteristics. However, these are not presented in the results section. They are mentioned only in the first paragraph of the discussion. As this is one of the key elemets to understand the results (and the lower success rate), I recommend adding a paragraph /table on baseline characteristics in the results section. As no propensity analysis or multivariate analysis have been performed one can not rule out the precise reason (CKD vs other baseline condition) for worse outcomes (i.e. mortality).

Responses: We added Table 3 and compare age, sex, LV function, J-CTO, contrast use between CKD group and non-CKD group according to this comment.

Comment 2: In the abstract, it is also mentioned the more complex lesions treated in the CKD group but no data are presented in this regard.

Responses: We have revised our manuscript according to this comment and added Table 3 to compare mean J-CTO score between CKD group and non-CKD group.

Comment 3: The last sentence of the abstract is not a conclusion, rather an objective of the study. Please rephrase the conclusions.

Responses: We have revised our manuscript according to this comment and revised one paragraph in Abstract on page 1, paragraph 1, lines 28-32, and in Conclusion on page 8, paragraph 1, lines 227-230, of the revised manuscript:

“In CKD CTO PCI population, a lower procedural success rate, a higher incidence of CIN or AKI and, higher in-hospital and long-term mortality rate were noted due to more complex lesions and more comorbidities. However, the incidence of TLR/TVR did not differ between groups.”

Comment 4: Data on outcomes in successful vs unsuccessful PCI are not presented. Please add this in the limitations section. 

Responses: We have revised our manuscript according to this comment and revised one paragraph in limitations on page 7-8, paragraph 1, lines 219-220,

“Fourth, no data about outcomes in successful versus unsuccessful PCI in such comparison were presented.”

Thank you for your constructive and valuable comments.

Round 2

Reviewer 1 Report

Dear author,

I reviewed the paper for a second time. 

Thank you for adapting the paper according to remarks. 

YS

JD

Reviewer 2 Report

All issues raised by this reviewer have been successfully addressed by the authors and the paer improved accordingly.